# The Onset of Depression in Middle-Aged Presumed Healthy Slovenian Family Practice Attendees and Its Associations with Genetic Risk Assessment, Quality of Life and Health Status: A Contribution for Family Medicine Practitioners’ Early Detection

**DOI:** 10.3390/ijerph18158197

**Published:** 2021-08-03

**Authors:** Nina Jerala, Polona Selič-Zupančič

**Affiliations:** Faculty of Medicine, Department of Family Medicine, University of Ljubljana, 1000 Ljubljana, Slovenia; jerala.nina@gmail.com

**Keywords:** depression, mental health, health-related quality of life, family practice, middle-age, comorbidities, genetic risk, primary prevention

## Abstract

Despite depression being a major driver of morbidity and mortality, the majority of primary care patients remain undiagnosed, so this study aimed to assess the prevalence of depression and the association with demographic and clinical variables, genetic risk, and quality of life. The participants were presumably healthy model family medicine practice (MFMP) attendees between 30 and 65 years of age and recruited during a preventive check-up in 2019. Each of the 40 pre-selected MFMP pragmatically invited 30 attendees to voluntarily participate. They completed a questionnaire of demographic, clinical, and social determinants, as well as a three-generational family history. The results were analyzed using multivariable modelling to calculate the associations with signs of depression. A modified Scheuner method was used to calculate the level genetic risk level using family history. Of 968 participants, aged 42.8 ± 8.6 years, 627 (64.8%) were women. The prevalence of depression was 4.1%. Signs of depression were negatively associated with health-related quality of life score, in particular in the domains of self-care (*p* = 0.001) and anxiety/depression (*p* < 0.001). Depression was also associated with predicted high risk for comorbidities given the family history (*p* = 0.030). Primary care directed at improving patients’ quality of life should implement more widespread screening for mental health disorders. Family history for disease even beyond depression can be used by physicians as an important primary prevention tool.

## 1. Introduction

Depression is a complex disorder that arises as a result of compounded genetic and environmental factors [1]. Mental disorders are a major driver of overall morbidity and mortality. Depression significantly negatively impacts the quality of life (QOL) [2,3], which is an important medical goal for patients suffering from chronic and incurable diseases, as well as essential for improving long term health outcomes and minimizing disability [3,4]. Health-related quality of life (HRQOL) is a related concept that measures the effects of health on five domains of life, including mobility, self-care, usual activities, mental health (anxiety/depression), and freedom from bodily pain [5]. Depression and other mental illnesses may cause distress and disability in many areas of life—via limitations in cognitive function, motivation, emotional regulation, social perception, and sense of self- and would thus be expected to cause a significant impact on HRQOL [2]. Additionally, depressed patients have a much higher mortality rate when compared to non-depressed patients [6], and worse physical health outcomes [7].

Previous studies have found that the lifetime prevalence of depression varies substantially across countries and has been shown to be anywhere from below 1% to up to 20% [8]. Studies looking at the Slovenian population have been similarly variable, estimating the prevalence between 3.4% and almost 25% in different groups [2,9,10,11,12,13,14], while the National Institute of Public Health estimates the overall prevalence in Slovenia to be 5.5% [15].

Studies looking at the population in general have estimated a prevalence of either 3.4% [9] or 9.1% [10]. Among family practice patients, depression is estimated to occur in around 10% worldwide [11], but there has been significant variability in prevalence between studies looking at Slovenian adult family practice attendees. Different studies have estimated the prevalence to be either around 6% [11], 13.5% [12], 15.2% [13], 15.5% [2], or 24.3% [14], though some of this variability could be ascribed to differing methods of assessment and execution within different periods. While the diagnosis of depression has become increasingly common in the last decades due to better-qualified family practitioners, increased awareness, and advancing treatment [16], it is estimated that two-thirds of individuals with depression are undiagnosed and untreated in primary care settings [17].

Many factors have been strongly and repeatedly associated with depression: female gender [18,19,20]; older age [19,21]; lower education [12,19,22]; lower socioeconomic status [19,23,24] and unemployment [24]; perceived social isolation [25,26]; alcohol consumption, drug abuse [27]; different stress-inducing situations [9,27]; and disruption to intimate relationships, such as divorce, spousal substance dependency, and intimate partner violence [12,14,28,29]. Between a third and a half of depression variance can be attributed to heritable factors [30], thus a positive family history is considered one of the three most powerful predictors of depression, together with recent major life events, and a personal history of past depressive episodes [31]. Family history of depression can be a risk factor for descendants even beyond two generations [32]. Depression has a complex, bidirectional connection to chronic somatic disease, as chronic disease patients are predisposed to depression and also experience more negative outcomes of their somatic disease due to depression [7,13,33,34,35]. Among somatic diseases, depression is especially intimately related to cardiovascular disease [36], but the evidence for association of individual contributing factors, such as blood pressure and lipid profile, is sparse and inconsistent [37,38,39,40,41]. As for other lifestyle factors, some studies have found that a healthy dietary pattern was significantly inversely associated with depressive symptoms [19,42,43]. The same was true for physical exercise [44]. An association between obesity and depression has also previously been established [45].

The aim of this study was to assess the prevalence of hidden depression in presumably healthy middle-aged population, and to explore the associations between depression and demographic characteristics, family history, lifestyle data, and clinical variables using multivariable modelling. Namely, during the study, high genetic risk for monogenetic and multifactorial diseases with an important genetic component was determined solely by an algorithm developed for this purpose. A three-generational family history to estimate the possibility of high genetic risk was used as an unspecific risk factor in this study.

As it has been estimated that 80% of people with major depressive disorder are treated entirely in primary care setting [46] and in the rest primary care is the most common entry point, it is of the highest importance that physicians have the tools to adequately screen, estimate risk, and understand the important factors in providing quality care for this highly prevalent condition.

## 2. Materials and Methods

### 2.1. Participants and Procedure

This study was conducted in the framework of a research project Development of an Algorithm for Determining of Genetic Risk at the Primary Healthcare Level: A New Tool for Primary Prevention (ID L7-9414) [32]. The participants were presumably healthy family practice attendees between 30 and 65 years, visiting model family medicine practices (MFMP) for a preventative check-up. MFMPs included in the study were 40 teams consisting of family practice physician, a practice nurse and a registered nurse, pre-selected trough pragmatic sampling and participating voluntarily after attending a short workshop to familiarize themselves with the methodology. Each team pragmatically included 30 consecutive patients, who participated voluntarily and signed an informed consent statement. The inclusion criteria were the age between 30 and 65, absence of chronic disease, and informed consent to participation. Exclusion criteria were age below 30 or above 65, and an inability to participate in the study—i.e., blindness, intellectual disability, psycho-organic impairment.

### 2.2. Instruments and Measures

The participants completed a questionnaire covering demographic data (age, gender) and a three-generational family history for monogenetic diseases and diseases with a genetic component, i.e., cardiovascular disease, hypertension, diabetes, cancer, neurological, mental and sensory issues, and other diseases present in the family with a possible genetic component. A web application-based algorithm with an aim for determining the risk level for selected monogenic and polygenic diseases was developed as part of a research project, “The Development of an Algorithm for Determining Genetic Risk at the Primary Healthcare Level: A New Tool for Primary Prevention” (ID L7-9414) [32]. The first part captures personal data (gender, number of abortions if women, number of siblings, having a twin, height, weight, presence of the current diseases), second section data on the structure and size of the three-generation family, third section captures data on the relatives’ diseases and the fourth section description of the risk of a familial genetic predisposition. There are no public health tools or electronic applications available to doctors or the lay public in Slovenia to determine the risk level for specific diseases with genetic etiology. The application [32] determines genetic risk for all diseases that could have a genetic component, not just for specific ones.

As part of the MFMP protocol, nutritional history, history of physical activity, smoking habits, alcohol consumption, self-assessed perception of stress, and signs of depression were collected, together with social health determinants.

The patients’ nutritional history was assessed using a 4-part questionnaire about a number of daily meals, vegetable and fat consumption, and use of salt. The points were awarded according to daily intake of 3–5 meals, regular breakfast, vegetables, fruit consumption, use of dairy products and milk with lower fat content, eating little or no processed meat products; not adding more salt to prepared food; using an appropriate type of fat for cooking and using an appropriate type bread spread. Nutrition was marked as appropriate with ≥14, satisfactory with 9 to 13 and inappropriate with ≤8 points appointed.

A participant was considered to be ‘borderline’ physically active when they were engaged in high-intensity physical activity twice per week, or in moderate-intensity physical activity 2–4 times per week. If a participant was less physically active than that, their activity was marked as ‘inadequate’, while if a participant was more physically active than that, their activity was marked as ‘adequate’.

Stress perception risk was derived from participants’ self-assessed stress load (not feeling under stress at all—feeling under stress every day) and ability to cope (successful coping—not coping at all) which both had 1–5 points and the sum 2–10; when ≥8, the high risk was appointed.

A two-question scale was used to identify signs of depression (how often the participant felt disinterested or dissatisfied with their daily activities in the past two weeks and how often they felt gloomy, depressed or desperate over the same period of time) each with up to 3 points and the answers weighted according to frequency (never—almost every day); the sum was up to 10 and the threshold ≥ 2 points.

Data concerning body mass index, blood pressure values, lipidogram, and blood sugar values was also collected. Cardiovascular risk was determined based on Framingham Risk Scores, and health related quality of life (HRQOL) was determined using the EQ-5D scale. The Slovenian version of AUDIT-C questionnaire was used to assess alcohol consumption.

### 2.3. Data Analysis

The data analysis for the whole research project has already been described elsewhere [32]. An algorithm based on the modified Scheuner method [47] was developed to determine the genetic risk of cardiovascular diseases, hypertension, diabetes, cancer, neurological, mental, and sensory diseases, and other diseases present in the family with a possible genetic etiology or component. The risk-assessment software has an implemented algorithm, which calculates the likelihood that a person will develop a hereditary disease based on the presence of medical conditions in their relatives. The application calculated the risk for diseases with a genetic component based on a three-generation family history. It classified the subjects in three risk groups: low, medium, and high. The risk level is based on the generation of the relatives affected by the disease, the onset of the disease in the relatives, and the number of relatives affected [32].

Using IBM SPSS 25 (IBM Corp., Armonk, NY, USA) software, we analyzed the data relating to demographic information, nutrition, smoking, alcohol consumption, level of physical activity, perception of stress, social health determinants, quality of life, cardiovascular risk, body mass index, blood pressure values, lipidogram and fasting blood glucose values, using multivariable binary logistic regression modelling to calculate the associations with signs of depression. We set the value of *p* < 0.05 as statistically significant.

### 2.4. Statistical Power of the Study

Total sample size was divided by the variance inflation factor (VIF) to address the multicollinearity among predictors included to regression modelling. The VIF, which depends on the squared multiple correlation coefficient (R^2^) relating a specific predictor of interest (in present study the high genetic risk) to the remaining predictors, was calculated according to the instructions in Hsieh [48]. We applied logistic regression and calculated Nagelkerke R^2^ for high genetic risk (R^2^ = 0.088). The effective sample size was reduced to 880 cases (total sample size of 968 divided by the VIF = 1/(1 − R^2^) = 1.10). Following Hsieh’s sample size tables [48], the effective total of 880 cases provides more than 95% power to detect a significant association for logistic regression (using an alpha of 0.05, a medium odds ratio of about 2.5 to 1 [49] and an event proportion of at least 4%).

## 3. Results

The questionnaire and the check-up were completed by 968 people. A third of those were men (35.2%; *n* = 341). The mean age was 42.8 ± 8.6 years (from 30 to 66), the mean EQ-5D score was 0.92 ± 0.15 (0.05–1.00), and the mean visual analogue scale (VAS) score was 84.5 ± 12.0 (min 30–100 max).

Signs of depression were shown by 4.1% of participants. The vast majority of participants (92.1%) was marked adequately or borderline physically active. While the majority of the participants consume alcohol (only 5.5% were marked as abstaining), most (90.4%) were non-risky drinkers. Similarly, the great majority (86.6%) were non-smokers. In the Framingham cardiovascular score, 63.8% were marked below 5% and 4.1% were marked high, between 20% and 40%. Interestingly, more than half (63.3%) had had inappropriate nutrition habits, with only 6.9% marked as appropriate, and the rest marked satisfactory. Still, almost half (49.5%) had an appropriate BMI (below 25). Most (96.4%) also had a low risk given the perception of stress. For laboratory values, 12.4% had a high blood pressure, and only 0.7% had a high glucose level. For the lipidogram, the scores were 23.2% for high cholesterol, 10.4% for high triglycerides, 11.6% for low HDL, and 64.3% for high LDL. In assessing HRQOL domains, 8.1% had problems with mobility, 0.7% with self-care, 23.3% with usual activities, 9.7% with pain or discomfort, and 12.1% with anxiety or depression. Using the Scheuner method, 35.8% (*n* = 347) were marked as high risk for genetic disease or a disease with a genetic component.

In the bivariate analysis, genetic risk of cancer was shown as statistically significant (*p* = 0.011) in the subgroups of analyzed diseases. Of all participants, 121 (12.5%) were shown as high risk; 11 of those had signs of depression (representing 27.5% of depressed group), and 110 did not (representing 11.9% of non-depressed group). Age was also shown as significant (*p* = 0.028), with the mean being 45.8 ± 8.8 in depressed group, compared to 42.7 ± 8.6 in non-depressed and 42.8 ± 8.6 in all participants. Depression was also negatively associated with each of the domains of HRQOL (*p* < 0.001).

With multivariable modelling, we were also able to negatively associate signs of depression with EQ-5D scores (*p* < 0.001). In particular, depression was associated with the domains of self-care (*p* = 0.001) and anxiety/depression (*p* < 0.001). Depression was also associated with predicted high risk for comorbidities given the family history (*p* = 0.030). The rest of the variables were not associated with signs of depression using multivariable modelling (*p* > 0.05).

## 4. Discussion

In this study, we successfully demonstrated an association between self-assessed depression, health related quality of life, and general genetic makeup as shown by three-generation family history.

The 4.1% prevalence of depression in this study is similar to the estimates of the National Institute of Public Health, which found 5.5% overall prevalence. In the subgroups within the age limits of this study (between 30 and 65 years old), they found a prevalence of 3.8%, 5.6%, and 4.8% for ages between 35 and 44, 45 and 54, 55 and 64 years, respectively [15]. However, compared to the other studies addressing depression prevalence in the Slovenian population, the number was unusually low, as they have generally found the prevalence to be in the proximity of 15% with 3.4% and 24.3% being the furthest outliers [2,9,10,11,12,13,14]. Though a part of this discrepancy might be explained by a lower age of participants in this study (42.8 yrs. (Table 1) vs. 44.2–53.1 yrs. [2,12,13,14]), the biggest factors was likely the recruitment of presumed healthy participants coming to the doctor merely for a preventative check-up, and the inclusion criteria of lack of chronic disease. Depression is decidedly associated with older patients with more comorbidities [13,19,21,24], a factor that is especially important in light of population aging and rise of multi-morbid patients [50]. Studies show that 20% of patients with chronic somatic disease also suffer from depression [13], which has a negative effect on the outcome of the somatic disease; at the same time, the somatic disease negatively affects depression [33,34,35].

Whereas most Slovenian studies recruited participants coming to family medicine doctors for health-related reasons and had a higher or even no upper age criterion, the study with the most similar results of 3.4% prevalence consisted of presumably healthy adults completing their regular occupational health check-ups and set the upper age at similar limit this study (64 and 65 years of age) [9]. They attributed the unexpectedly low prevalence to the protective effect of employment, but while the study did not account for the prevalence of chronic disease in the sample, it might be assumed that the younger working population they studied (mean age 40.5) had a lower prevalence of chronic diseases. We might therefore consider the possibility that the major factors for the results of both studies was the lower age and the lack of comorbidities.

A high risk for a genetic disease or a disease with a genetic component (cardiovascular disease, hypertension, diabetes, cancer, neurological, mental and sensory issues, and other diseases present in the family with possible genetic etiology or component) was shown to be a non-specific risk factor for signs of depression in this study (Table 2).

It has long been known that family history is one of the most important predictive tools for the diagnosis of depression, as twin studies suggest between 33% and 46% of the variance is due to inherited genetic factors [30]. An array of studies suggests a preeminent role of genetic alterations of the three major monoamine systems (serotonin, norepinephrine, and dopamine circuits), with more recent conceptual approach involving several critical brain regions and associated pathways, as well as other structural, neuroendocrine, and immunologic changes in the body, influenced by epigenetic mechanisms and the environment [51]. Early studies used a number of approaches and faced many difficulties identifying any specific inherited factors, most likely due to depression being complex and vastly heterogeneous, with a great number of genetic loci, each contributing only a small effect [52]. However, Wray et al. presented recent much larger genome-wide association meta-analyses which successfully identified a number common genetic variants in people with depression, with notable reported genes including genes associated with presynaptic differentiation and vesicle trafficking, signaling, including neuronal calcium signaling, dopaminergic and glutamate neurotransmission, hypothalamic–pituitary–adrenal axis activation, major histocompatibility complex, and neuroinflamation [52]. Several of these studies have independently provided evidence of the importance of cortical brain regions via enrichment analysis and brain imaging, with proven associations the identified genes and many other diseases and traits with a genetic component, including many psychiatric disorders, neuroticism, sleeping patterns, education level and female reproductive life events [52,53]. Other complex chronic diseases have also been identified and a number of them are associated with depression as well. Predisposition for diabetes, for example, is considered complexly inherited [54], but diabetes is also known to be significantly more prevalent in depressed and anxious patients [12]. Depression may be considered an independent prognostic factor for coronary heart disease [14] and significant positive correlation has also been found in calculating genetic correlations between depression and coronary artery disease in aforementioned genome-wide association meta-analyses [52,53]. Certain types of malignancies, such as colon cancer, have been shown to be predicted by depression in prospective population cohort studies [14], and are known to be heritable [32]. Other complex and somewhat heritable diseases have also been found to be positively correlated with depression, including rheumatic disease, asthma, and migraine [12,14].

It has long been assumed that an important reason for the association between chronic disease and depression is the burden of symptoms, chronic pain, and disability related to many chronic somatic diseases. This study, however, suggests that there might be a possibility of coinheritance, or that the complex interplay of genetic factors in heterogeneously inherited diseases may contribute to the development of more than one disease. As the genetic associations for major depression tend to occur in genomic regions conserved in many placental mammals, it seems that these genes have important functional roles, which might also contribute to the possibility of the genes being related to a number of conditions [52]. We further theorize that highly symptomatic disease, such as cancer, experienced by a child’s primary caregiver, contribute to the parent not being able to provide continuous care, and could be considered as a stressful or traumatic childhood event, also predisposing an individual to develop depression later in life [51]. Indeed, cancer was shown as a particularly significant disease group in the bivariate analysis (*p* = 0.011; table not shown).

In any case, family history for disease even beyond depression can be used by physicians as an important primary prevention tool in healthy individuals, helping them identify and administer additional screening for people at increased risk for depression. While typically a family history is used to trace the origins of a disease in patients already afflicted, a more novel approach utilitizes establishing the risk of predisposition to a disease even in healthy people, thus enabling early detection or even prevention of the disease, which would lead to a higher quality of life. QOL refers to the degree to which an individual is healthy, comfortable, and able to participate in or enjoy life events. Health related quality of life (HRQOL) is a related concept, which embraces the concepts of functional status, health status, quality of life, and well-being in relation to the individual’s health [5].

While in the light of other studies a prevalence of 4.1% seems low, it is nonetheless notable that 1 in 25 of by all accounts healthy individuals suffers from unrecognized depression, especially considering the effect of this condition on the individuals’ quality of life. Studies suggest that even subthreshold symptoms of depression and dysthymia are associated with a small, but significant reduction in quality of life [55,56], which is in concordance with the results we show here. Many studies have shown that the burden of mental disorders, particularly mood disorders, accounts for a substantial impairment in HRQOL, which tends to far exceed the impairment from chronic somatic illness and disability [2,5,14]. Previous studies in the Slovenian population confirm this observation, as they found that the burden of somatic comorbidity was smaller than the impact of psychosocial determinants [14]. Therefore, we might consider that mental disorders and the related HRQOL might be the most important determinant in how primary care patients feel about themselves, their lives, and the healthcare they receive, as has been suggested by other authors before [5]. HRQOL is thus increasingly being recognized as an important indicator of primary care quality, and studies reveal that patients would in fact like to have their functional performance and emotional well-being assessed as a part of their medical care [5].

While other studies have found that mood disorders in particular are associated with decrements in all domains in HRQOL [5], we have found specific associations with the domains of anxiety/depression and self-care. As anxiety/depression is the specific component of HRQOL that pertains to and takes into account mental health, it is only natural that it would be affected. The decrease in the domain of self-care, while not directly related to mood disorders, is consistent with the literature as well, as depressed individuals often have significant impairments in functioning as relating to self-care, work, productivity, and social and physical aspects of life [56].

Previous research in Slovenia has shown that intimate partner violence (IPV) is a factor closely associated with depression and HRQOL. One study found that 37% of participants affected by any exposure to IPV at any point in their life also showed signs of depression, a rate twice higher than in participants with chronic pain or other conditions [28]. A different study in the same environment detected an 89.5% rate of depression chronic and general anxiety disorder in the participants exposed to IPV in the previous year [29]. This should be considered by the local family doctors as lifetime IPV exposure was found to be 40.4% and has been experienced by as much as 36.9% of the individuals with depression in the study, which is consistent with other research. Other authors have also suggested that inquiry into depression, possible abuse, and dysfunctional family dynamics should be added to a typical clinical evaluation [28].

Notably, in a longitudinal study on a representative sample of Slovenian family medicine patients, the factors consistently positively associated with the mental component of QOL were the very antithesis of IPV, namely social support, satisfactory circumstances in patients’ household, and absence of anxiety [2].

We were not able to show an association for some factors that have previously been established as associated with depression. For instance, we did not show an association with older age (Table 2) that has previously been well established [19,21]. Our sample consisted of presumed healthy individuals between 30 and 65, so it is possible that the range of years was not sufficient to notice an important divergence, especially since the participants by design did not suffer from any comorbidities and the upper age might have been too low to reach the more commonly afflicted elderly population. We were also not able to show a previously established association with risky drinking and perceived stress level [9,27], which we attribute to a relatively rare occurrence of these determinants in the sample (4.1% and 3.6% respectively; Table 2). Combined with the relatively low prevalence of depression the statistical power might have been too low to show an association in these subgroups. There is limited evidence in the literature that a healthy diet is associated with a lower risk of depression [19,42,43], but as ‘healthy diet’ is a loosely defined and inconsistent term, it is possible that our methods of assessing adequacy of nutrition (number of meals, amount of salts and fats in the diet) were not sufficient to make an association. While we were not able to show an association with biochemical values of blood pressure and lipidogram, the literature on the topic is limited and inconsistent at best. For blood pressure, studies have found either a positive, negative, or non-existent relationship to depression [37]. Similarly, studies have found variable associations between depression and lipid profile, with positive association for lower total cholesterol [38,39], lower HDL [40], lower LDL [38], lower non-HDL cholesterol [38], or even higher cholesterol in the case of younger individuals [41]. Age has previously been shown to be a risk factor [19,21], but in this study it was only shown as significant in bivariate analysis (*p* = 0.028; table not shown), but not in multivariable modeling.

### Strengths and Limitations to the Study

A major strength of this study is the large sample size, which allows the results to be generalized for the middle-aged Slovenian population and thus provides valid proof to the hypothesis that seemingly healthy individuals might still exhibit less obvious signs of disease, such as depression, and might therefore greatly benefit from targeted interventions by their physicians. It should be noted, however, that two thirds of the participants were women, who are already more susceptible to depression, so the results should be observed with this in mind. An additional considerable strength was the splitting of the EQ-5D score into five domains and thus establishing the association with individual areas—i.e., mobility, self-care, usual activities, pain/discomfort, and anxiety. Finally, the validation of inclusion of positive family history as a risk-forecasting tool in presumably heathy population represents another means of targeted intervention by the physician.

It should be noted that some of the data (nutritional history, physical activity, smoking, alcohol consumption, perception of stress, QOL) was not corroborated with objective measurements, but merely self-reported, and is therefore subjective to the participants’ memories and experiences. While the sample was large, a relatively small percentage of participants with signs of depression means a lower prediction power in this subgroup, so some expected associations could not be confirmed by statistical analysis. While the healthcare providers were not given any specific additional education about the investigated determinants in order to interfere as little as possible with the real-life approach of the study, the nurses collecting the data were aware of the patient’s voluntary participation in the study and might have unconsciously altered their usual care patterns. Additionally, just the fact of participation could have altered the patient’s mood and perception of their health, as it might have given them an additional sense of safety. Given the subjective nature of the assessment of HRQOL it should be noted that our findings could be somewhat biased by the possibility that people with mood disorders might offer an overly pessimistic self-appraisal of their state. We determined the presence of depression using only two questions, which some might consider to be insufficient to achieve adequate specificity and sensitivity, however, studies show that a single question on depressed mood can detect 85–90% of patients with major depressive disorder [46]. Therefore, we consider this to rather be a strength of this study, as this method can very quickly and conveniently be used by any physician at any visit.

## 5. Conclusions

A high risk for genetic disease or a disease with a genetic component (cardiovascular disease, hypertension, diabetes, cancer, neurological, mental and sensory issues, and other diseases present in the family with possible genetic etiology or component) as calculated using three-generational family history was shown to be a non-specific risk factor for signs of depression, which physicians could use as a primary prevention tool to prevent or alter the course of the disease in at risk patients, leading to a higher quality of life, given the association between HRQOL and depression that was also shown in this study. Signs of depression were specifically associated with the depression/anxiety and self-care domains of HRQOL. As quality of life is perhaps the most important factor for the patient and an important indicator of the quality of primary care, primary care directed at improving HRQOL should implement more widespread screening for mental health disorders, which can be accomplished in as much as two questions.

## Figures and Tables

**Table 1 ijerph-18-08197-t001:** Characteristics of participants.

	*n* = 968	%
Gender		
Female	627	64.8
Male	341	35.2
Signs of depression		
No	928	95.9
Yes	40	4.1
Physical activity		
Inadequate	76	7.9
Adequate/borderline	892	92.1
Alcohol consumption status		
Abstinent	53	5.5
Non-risky drinking	875	90.4
Risky drinking	40	4.1
Smoking		
No	838	86.6
Yes	130	13.4
Framingham cardiovascular risk score		
<5%	618	63.8
5–10%	197	20.4
10–20%	113	11.7
20–40%	40	4.1
Nutrition		
Inappropriate	613	63.3
Satisfactory	288	29.8
Appropriate	67	6.9
Perception of stress		
Low risk	933	96.4
High risk	35	3.6
BMI (kg/m^2^)		
<25	479	49.5
<30	367	37.9
>30	132	13.6
High blood pressure (>140/90 mmHg)	120	12.4
High blood glucose (>7 mmol/L)	7	0.7
High serum cholesterol (>6 mmol/L)	225	23.2
High serum triglycerides (>2 mmol/L)	101	10.4
Low serum HDL (<1.2 mmol/L men; <1.0 mmol/L women)	112	11.6
High serum LDL (>3 mmol/L)	622	64.3
High genetic risk	347	35.8
Eq1 mobility		
No problems	890	91.9
Any problems	78	8.1
Eq2 self-care		
No problems	961	99.3
Any problems	7	0.7
Eq3 usual activities		
No problems	742	76.7
Any problems	226	23.3
Eq4 pain/discomfort		
No problems	874	90.3
Any problems	94	9.7
Eq5 anxiety/depression		
No problems	851	87.9
Any problems	117	12.1
	M ± SD	range
Age (years)	42.8 ± 8.6	30–66
Eq5d TTO	0.92 ± 0.15	0.05–1.00
VAS	84.5 ± 12.0	30–100

M: mean, SD: standard deviation.

**Table 2 ijerph-18-08197-t002:** Associations between demographic values, health status, genetic risk, and depression.

	OR (95% CI)	*p*
Age (per 10-years increase)	1.36 (0.82–2.27)	0.238
Gender		
Male	1.00 (reference)	
Female	1.23 (0.49–3.08)	0.652
Physical activity		
Inadequate	1.00 (reference)	
Adequate/borderline	0.85 (0.27–2.68)	0.776
Alcohol drinking status		
Abstinent	1.00 (reference)	
Non-risky drinking	1.35 (0.26–6.93)	0.717
Risky drinking	0.65 (0.04–9.98)	0.756
Smoking		
No	1.00 (reference)	
Yes	1.75 (0.66–4.66)	0.265
Nutrition		
Inappropriate	1.00 (reference)	
Satisfactory	1.41 (0.63–3.18)	0.402
Appropriate	2.37 (0.70–8.01)	0.166
Perception of stress		
Low risk	1.00 (reference)	
High risk	0.71 (0.14–3.62)	0.682
Framingham cardiovascular risk score		
<5%	1.00 (reference)	
5–10%	0.46 (0.16–1.33)	0.152
10–20%	0.90 (0.26–3.08)	0.865
20–40%	1.04 (0.31–3.65)	0.927
BMI (kg/m^2^)		
<25	1.00 (reference)	
<30	0.82 (0.36–1.91)	0.654
>30	0.45 (0.13–1.59)	0.213
Eq1 mobility		
No problems	1.00 (reference)	
Any problems	1.54 (0.51–4.63)	0.445
Eq2 self-care		
No problems	1.00 (reference)	
Any problems	6.08 (2.29–16.15)	0.001
Eq3 usual activities		
No problems	1.00 (reference)	
Any problems	2.37 (0.97–5.80)	0.059
Eq4 pain/discomfort		
No problems	1.00 (reference)	
Any problems	0.88 (0.31–2.51)	0.814
Eq5 anxiety/depression		
No problems	1.00 (reference)	
Any problems	9.69 (4.29–21.89)	<0.001
High genetic risk		
No	1.00 (reference)	
Yes	2.34 (1.09–5.03)	0.030

OR: odds ratio, CI: confidence interval, Nagelkerke R^2^ = 0,318.

## Data Availability

The raw data presented in this study are available on request from the corresponding author. The data are not publicly available due to policy of institution that gave ethical approval to the study.

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
