# Peer review of "The Onset of Depression in Middle-Aged Presumed Healthy Slovenian Family Practice Attendees and Its Associations with Genetic Risk Assessment, Quality of Life and Health Status: A Contribution for Family Medicine Practitioners’ Early Detection"

_ijerph, 2021, doi:10.3390/ijerph18158197_

Round 1

Reviewer 1 Report

Dear authors,

You prepared an interesting and important article about a new tool for use in the primary care settings. I have some minor comments and questions.

  1. Line 276 - I miss the reference
  2. Line 290 - I miss the reference
  3. Line 320-334 - could you please add some explanation why and how have you find a connection between your research and the IPV (it is not mentioned in the methodology that you have had researched that topic)
  4. Line 361 - I can't agree that the results could be generalized for Slovenian population. You have included the participants from 30-65 years, so you could generalise the results only for the middle aged group like you already mentioned it at the title of the article. 20,7% of the Slovenian population is older then 65 years.

With kindly regards, reviver 2

Author Response

Dear Reviewer #1,

Thank you for your thorough review and valuable comments.

We have amended the article as follows:

Point 1: Line 276 - I miss the reference

Response 1: On further review, we noticed this sentence was a misunderstanding of a claim in the reference cited. We have revised the sentence and provided a new reference.

Please see lines 277-280.

Point 2: Line 290 - I miss the reference

Response 2: The reference was checked, it is #51. Afterwards we concluded that the R#1 was not satisfied with our presentation of the content, i.e. ''an inconsistent parental response or lack of atonement for a child’s needs in early life'' and make the wording in our summarisation more clear:

We further theorize that highly symptomatic disease, such as cancer, experienced by a child’s primary caregiver, contribute to the parent not being able to provide continuous care, and could be considered as a stressful or traumatic childhood event, also predisposing an individual to develop depression later in life [51].

Please see lines 294-296.

Point 3: Line 320-334 - could you please add some explanation why and how have you find a connection between your research and the IPV (it is not mentioned in the methodology that you have had researched that topic)

Response 3: The research group these authors belong to has previously studied the association between IPV and depression. Given the high co-occurrence of the two in Slovenia, we felt it necessary to elaborate on IPV in the discussion section and referenced this previous finding.

Please see lines 335-336.

Point 4: Line 361 - I can't agree that the results could be generalized for Slovenian population. You have included the participants from 30-65 years, so you could generalise the results only for the middle aged group like you already mentioned it at the title of the article. 20,7% of the Slovenian population is older than 65 years.

Response 4: We have modified the wording as suggested, generalising the results only for the middle aged group.

Please see line 376.

(Also added in attachment)

Reviewer 2 Report

I found one error in line 227 »healthy related quality of life«- please correct the typo; there should be “health related quality of life«

Further on I have some minor suggestions:

- please change multivariable logistic regression modelling to multivariable    binary logistic regression modelling
- please change expression multivariate to multivariable
- please in case of R2, digit should always be superscript
- please add description of bivariate tests to data analysis section

Author Response

Dear Reviewer #2,

We are grateful for your suggestions and modified the text as follows:

Point 1: I found one error in line 227 »healthy related quality of life«- please correct the typo; there should be “health related quality of life«

Response 1: We have corrected the typo.

Please see line 221.

Point 2: Please change multivariable logistic regression modelling to multivariable binary logistic regression modelling

Response 2: We changed the wording.

Please see line 168

Point 3: Please change expression multivariate to multivariable

Response 3: We changed the wording.

Please see throughout the text and also the Abstract (line 17, 81, 168, 215, 373)

Point 4: Please in case of R2, digit should always be superscript

Response 4: We have corrected the digit.

Please see lines 173, 176, 177.

Point 5: Please add description of bivariate tests to data analysis section

Response 5: We believe that it is sufficient to add that the data analysis for the whole research project has already been described elsewhere [32]. If the Reviewer #2 finds it essential for the clarity of this text, we will provide an additional description.

Please see lines 153-154.

(Also added in attachment)
